# Warm Blood Meal Increases Digestion Rate and Milk Protein Production to Maximize Reproductive Output for the Tsetse Fly, *Glossina morsitans*

**DOI:** 10.3390/insects13110997

**Published:** 2022-10-31

**Authors:** Joshua B. Benoit, Chloé Lahondère, Geoffrey M. Attardo, Veronika Michalkova, Kennan Oyen, Yanyu Xiao, Serap Aksoy

**Affiliations:** 1Division of Epidemiology of Microbial Diseases, Yale School of Public Health, 60 College St., New Haven, CT 06510, USA; 2Department of Biological Sciences, University of Cincinnati, Cincinnati, OH 45221, USA; 3Department of Biochemistry, Virginia Polytechnic Institute and State University, Blacksburg, VA 24061, USA; 4The Fralin Life Science Institute, Virginia Polytechnic Institute and State University, Blacksburg, VA 24061, USA; 5Center of Emerging, Zoonotic and Arthropod-Borne Pathogens, Virginia Polytechnic Institute and State University, Blacksburg, VA 24061, USA; 6The Global Change Center, Virginia Polytechnic Institute and State University, Blacksburg, VA 24061, USA; 7Department of Entomology at Virginia Polytechnic Institute and State Univerity, Blacksburg, VA 24061, USA; 8Department of Entomology and Nematology, Division of Agriculture and Natural Resources, University of California Davis, Davis, CA 95616, USA; 9Section of Molecular and Applied Zoology, Institute of Zoology, Slovak Academy of Sciences, 814 38 Bratislava, Slovakia; 10Department of Mathematical Sciences, University of Cincinnati, Cincinnati, OH 45221, USA

**Keywords:** tsetse, digestion, thermal stress, reproduction

## Abstract

**Simple Summary:**

Consumption of a bloodmeal represents a stressful period for arthropod vectors. A general consensus is that thermal stress from a bloodmeal is detrimental. Here, we examined if the warm bloodmeal is critical to digestion and reproduction in tsetse flies. These results show that warm blood consumption is critical to reach maximum digestion efficiency. This increased digestion rate allows the flies to maximize the production of milk proteins. Importantly, modeling indicates that this consumption of warm blood contributes to increased population growth rates when compared to ingesting a cool bloodmeal. These studies are important as thermal impact of a warm bloodmeal can vary between beneficial or detrimental depending on the species arthropod vector.

**Abstract:**

The ingestion of blood represents a significant burden that immediately increases water, oxidative, and thermal stress, but provides a significant nutrient source to generate resources necessary for the development of progeny. Thermal stress has been assumed to solely be a negative byproduct that has to be alleviated to prevent stress. Here, we examined if the short thermal bouts incurred during a warm blood meal are beneficial to reproduction. To do so, we examined the duration of pregnancy and milk gland protein expression in the tsetse fly, *Glossina morsitans*, that consumed a warm or cool blood meal. We noted that an optimal temperature for blood ingestion yielded a reduction in the duration of pregnancy. This decline in the duration of pregnancy is due to increased rate of blood digestion when consuming warm blood. This increased digestion likely provided more energy that leads to increased expression of transcript for milk-associated proteins. The shorter duration of pregnancy is predicted to yield an increase in population growth compared to those that consume cool or above host temperatures. These studies provide evidence that consumption of a warm blood meal is likely beneficial for specific aspects of vector biology.

## 1. Introduction

Tsetse flies are one of the few insects that employ viviparity [1,2], characterized by the provision of nutrients beyond egg yolk to support embryonic development and growth of larva to full term within the female uterus. Female tsetse produce a single mature third instar larva during each gonotrophic cycle following a 4–6 day period of intrauterine gestation [1,3]. Thus, these K-strategists produce only a modest 8–10 progeny per female, per lifetime [1,3,4,5]. A critical adaptation underlying this reproductive strategy is the modification of the female accessory gland to secrete milk into the uterus for larval consumption. The milk is composed of proteins and lipids emulsified in an aqueous base [6,7]. During lactation, at least 6–10 mg of nutrients dissolved in 12–14 mg of water are transferred to the larva. Molecular characterization of tsetse milk revealed 12 major milk gland proteins, including Transferrin [7,8], a Lipocalin (Milk Gland Protein 1, MGP1 [7,9], nine tsetse-specific milk proteins (MGP2-10; [7,10,11]), and Acid Sphingomyelinase 1 (aSMase1; [12]). Transcriptomic analysis revealed that the milk proteins represent over 47% of the total transcriptional output in lactating flies. The contribution of the milk protein transcripts declines to less than 2% of the total output two days after the lactation cycle, at parturition [7]. This high investment in the progeny indicates that nutrients need to be rapidly processed for storage or direct allocation to the developing larvae. As an obligate blood feeding insect throughout their adult life, all nutrients are derived from the consumption of a blood meal. 

Previous studies have shown that the ingestion of a warm blood meal can be detrimental to blood feeding insects, where an increase in from ambient (24–26 °C) to the temperature of the host (37–40 °C) can cause biological damage [13,14,15,16]. For mosquitoes, heat shock proteins are increased during the blood meal, which if suppressed reduces egg production [14]. Species that urinate during feeding, including soft ticks and *Anopheles* mosquitoes will prevent the increase in thermal stress by evaporative cooling [13,14,15,16]. Kissing bugs have a countercurrent heat exchange system that prevents thermal damage from the bloodmeal, where inhibition of this mechanism yields an increase in the expression of heat shock proteins [15]. Tsetse flies are not immune to this process, as a drastic increase in body temperature occurs during blood feeding [17]. As tsetse flies ingest multiple bloodmeals to produce the nutrients necessary to complete larvogenesis, this suggests that the response to thermal stress during the bloodmeal may diverge to egg producing species that only need a single bloodmeal to reproduce. 

Importantly, as increased temperature is critical for the digestive process [18], a warm blood meal may facilitate rapid processing of the meal. This could subsequently result in increased reproductive output, which represents a critical factor allowing tsetse flies to maintain population levels [1,12,19,20,21,22]. In the present study, we examined the impact of blood meal temperature on pupal mass and duration of pregnancy. This was followed by the measurement of blood digestion, milk protein gene expression and modeling of these effects on population dynamics. These results indicate that the consumption of a warm blood meal is likely critical for tsetse flies to reach their maximum fecundity. 

## 2. Materials and Methods

### 2.1. Flies and Description of Studies

*Glossina morsitans morsitans* (Westwood, 1851) were reared at Yale University. Flies were maintained on bovine blood meals provided through an artificial feeding system at 48 h intervals [19]. The temperature of the feeding system was regulated to provide the described experimental treatments. The specific temperatures used were 30, 33, 36, 37, 38, and 41 °C, which varied between the experiments. Temperatures varied +/−1 °C from the listed temperatures. Flies were also allowed to consume cool blood early in the pregnancy cycle and warm blood later in the pregnancy cycle (30–38 °C). Flies that did not feed were removed from the study. Flies were used after the first pupal deposition, indicating that the time of pregnancy represents that second pregnancy cycle. Pupal mass was measured gravimetrically with a CAHN 21 electrobalance. Pregnancy duration and pupal mass were determined for 30 individuals for each treatment. All statistics were conducted with the use of R packages (3.6.3). 

### 2.2. Thermal Imaging and Contact Thermal Changes during Blood Meal

Thermal imaging was conducted as in Lahondère and Lazzari (2015). Briefly, a thermographic camera (PYROVIEW 380L compact, DIAS infrared GmbH, Germany; spectral band: 8–14 mm, uncooled detector 2D, 384 × 288 pixels) with a macro lens (pixel size 80 µm; A = 60 mm; 30° × 23°) was used for data acquisition [17]. The emissivity was set at 0.98 as determined previously for insect cuticle [20]. 

Direct temperature changes were monitored using a thermocouple (Omega) attached to the tsetse fly using petrolatum on the top of the abdomen based on similar methods used in mosquitoes [14], and tsetse flies were placed on an artificial feeding system as described previously. Temperatures of eight females were monitored during feeding with a HHM290 thermometer (Omega). 

### 2.3. Blood Digestion

Blood digestion quantification was determined according to previous methods [21]. Flies were offered blood at either 30 or 37 °C. Briefly, flies at five days post-parturition were dissected at four and sixteen hours following a blood meal to remove the digestive tract. The digestive system was removed from the proventriculus to the point of Malpighian tubules insertion. All fat body and tracheal tubes were removed. If the gut was ruptured during dissection, the sample was not used in further analyses. The guts and associated contents were dried at 50 °C in the presence of drierite (Xenia, OH, USA) until the mass was constant. The dry mass was set as the relative amount per 50 mg of blood to allow comparison to previous studies [22]. Eight individual guts were used for each treatment. 

### 2.4. RNA

RNA isolations were performed using Trizol reagent on whole flies following the instructions provided by the manufacturer (Invitrogen). RNA was cleaned with an RNeasy Mini Kit (Qiagen). Complementary DNA was synthesized using a Superscript III reverse transcriptase (Invitrogen) kit from 1 µg of the total RNA isolated from each sample. Flies were collected 12 h following the bloodmeal with blood digestion usually complete within 36–48 h, indicating that our samples represent the early digestion phase. 

### 2.5. Quantitative PCR

Transcript levels for *mgp1*, *asmase1*, *mgp7*, and *trf* (gene sequences acquired from the *Glossina* genome project, vectrobase.org) were determined via qPCR using the CFX real-time PCR detection system (Bio-Rad, Hercules, CA, USA) with primers specific to each target gene. Primers used in this study were used as in previous studies [7,12,22]. All readings were obtained on four biological replicates (two flies merged per replicate) normalized to tsetse *tubulin* expression levels. CFX Manager software version 3.1 (Biorad) was used to quantify the transcript abundance of each gene and conducted according to methods developed in previous studies [12,23]. All statistics were conducted with the use of R packages (3.6.3). 

### 2.6. Population Modeling

We developed a mathematical model to examine the population dynamics of tsetse flies using Matlab (Mathworks). We consider three compartments representing immature female (***P***), matured female within the first deposition period (***A*_1_**), and matured female in deposition periods following the initial deposition (***A*_2_**,). Here, the immature class is assumed to include both populations of larvae and pupae. We let $1/d_{p}$ and $1/d_{a}$ be the expected duration of the immature (larval and pupal development) and matured females (longevity). $a_{1}$ and $a_{2}$ are the development rates (combined larval and pupal development) from P to $A_{1}$ and $A_{1}$ to $A_{2}$, respectively. Our model then is followed as: $$\frac{dP}{dt}=B\left(temperature,\;A_{1},A_{2}\right)-\left(d_{p}+a_{1}\right)P,$$
$$\frac{dA_{1}}{dt}=a_{1}P-\left(d_{A}+a_{2}\right)A_{1},\;$$
$$\frac{dA_{2}}{dt}=a_{2}A_{1}-d_{A}A_{1},$$
where $B\left(temperature,\;A_{1},A_{2}\right)$ represents the temperature and density dependent birth rate. Based on the analysis in Hu et al. 2008 and Benoit et al. 2018, which describe the specific parameters such as longevity and pupal viability, we calculated the average expected number of female offspring of a female parent per unit time during each deposition as:$$r=ms_{p}\frac{d_{A}e^{-d_{A}t_{1}}}{\left(1-e^{-d_{A}t_{2}}\right)},\;$$
where m is the proportion of offspring that are female; $s_{p}\;$ is the survival rate of deposited larva; $t_{1}$ and $t_{2}$ are the duration of the previous and current deposition cycle.

During the first deposition, to calculate the average daily birth rate of the group $A_{1}$, we use $t_{1}=t_{2}+8$. This is because the duration of the first deposition is longer than that of other deposition cycles. To estimate *Glossina* populations after 1 year, with the initial population being 100 pupae and 100 adults, we calculated the average of the populations for 100 runs of the model simulations under different temperature settings. For each temperature setting and each run, we use rando-sampled life span for adult populations, and the durations of each deposition till the end of 1-year-period are also randomly sampled from our experiment dataset.

## 3. Results

### 3.1. Temperature Changes in Tsetse Flies Associated with a Blood Meal

Thermal imaging revealed that the flies’ body temperature increased to near the temperature of the ingested blood (37 °C) (Figure 1A). When the temperature was tracked with a thermoprobe when feeding on warm or cool blood, there was a much greater increase in temperature of the fly thorax when feeding on warm (37−38 °C) rather than cool blood (30−31 °C) (Figure 1B). These results indicate that a warm blood meal increases the flies’ body temperature more drastically than during the ingestion of a cool blood meal. 

### 3.2. Pupae Size and Duration of Pregnancy

When a blood meal was offered at multiple temperatures, no difference in the size of the pupae produced was found (Figure 2A). This is not surprising as few differences are noted in the size of pupae produced when larviposition occurs. When the duration of pregnancy was assessed, there was a significant reduction in the duration of pregnancy when flies were fed at 38 °C (F_5.178_ = 1.35, P = 0.018) (Figure 2B). This reduction only occurred at this temperature but was not observed when the blood was at 41 °C. This indicated that a warm, but not hot, blood meal may be beneficial to tsetse flies by reducing pregnancy duration.

### 3.3. Blood Processing and Milk Protein Synthesis

When the mass of the midgut was examined as a proxy for digestion, we noted that there was a delay in the processing of the blood meal (Figure 3) in flies that consumed a cool blood meal. After twelve hours, the dry mass of these flies was significantly higher than the individuals that consumed a warmer blood meal. The expression of milk gland protein genes was lower in the flies that were offered only cool blood meals before 6–7 days into a pregnancy cycle (Figure 3). The results suggest that ingesting a cool blood meal leads to a delay in the pregnancy cycle, likely due to a slower blood digestion and a delay/reduction in the production of milk proteins.

### 3.4. Population Modeling

To determine the impact of warm or cold blood meals on the population dynamics of *Glossina*, we used the model generated in this study to estimate the populations of tsetse flies after 1 year with the initial population consisting of 10 pupae and 10 adults within their first deposition (=birth) period. These parameters were based on previous studies [22,24]. We evaluated the average of the populations for 100 runs of the model simulations under different temperature settings. We determined that the average total population when feeding on warm blood (37–38 °C) after 1 year is significantly higher than those fed on blood at other temperatures even with small initial population (Figure 4a), but if individuals fed on a warm blood meal and then cool this benefit was eliminated (Figure 4b).

## 4. Discussion

In this study, we show that the temperature of a blood meal can impact the reproductive output of tsetse flies by reducing the overall duration of pregnancy. This decreased pregnancy cycle is likely due to an increased time necessary for blood meal processing, which yields a decreased production of milk proteins. This reduced pregnancy cycle likely yields a higher population growth when compared to flies that consumed either cool blood or blood at temperatures above that of a normal host. These results suggest that ingestion of warm blood is critical to tsetse fly reproduction. Tsetse flies will likely have most of their bloodmeals at near optimum temperatures, 36–38 °C, but variations can occur where some host species have higher blood temperatures (40–42 °C, [14,16]). Additionally, thermal variation within specific vertebrate host can lead to low blood temperatures (32–35 °C) continually, for specific body areas, and temporally [14,16,25,26,27,28,29]. This highlights that tsetse flies could potentially ingest blood across the temperature gradient assessed in the study.

Blood ingestion represents a period of extreme stress that includes increased oxidative stress [25], toxicity from the overabundance of specific amino acids and ions, and an overabundance of water and iron products. Blood feeding insects tolerate these stresses via a combination of adaptations developed over the course of their evolution into this life strategy. Specifically, several mechanisms allow insects and other blood feeding arthropods to remain cool during a blood meal. This includes a counter-current heat exchange that allows kissing bugs to remain cool while feeding [15] or evaporative cooling through urination that has been documented in mosquitoes and ticks [13,16]. Damage prevention at the cellular level is accomplished through the expression of heat shock proteins [14,15,16]. Suppression of these proteins impacts the processing of the blood and egg production. Thermal stress has been mainly assumed to be a detriment [15,16], but as digestion is directly tied to temperature in tsetse flies [18], this temperature increased could be a benefit. Based upon our results, digestion is delayed when cool blood is consumed, suggesting that increased temperature during the blood meal is critical for tsetse flies to rapidly process their blood meal. The viviparous biology of tsetse flies is unique and requires that multiple bloodmeals are acquired to complete pregnancy cycle. This process may necessitate a higher tolerance to repeated warm bloodmeals to maximize progeny output. Importantly, blood digestion could be delayed in oviparous species feeding on a cooler bloodmeal, but that has yet to be studied and as most species require a single bloodmeal each gonotrophic cycle may not be as critical.

The rapid processing and incorporation of the blood meal is critical for tsetse flies to reach their maximum reproductive output. Immediately after larval deposition, there is a rapid period of milk gland involution [22], which allows the female to begin to accumulate nutrient reserves. During this time, flies consume fewer blood meals, usually only two-three, that function directly as a source of energy and facilitate accumulation of lipid reserves [1,26,27]. Likely, the observed delay in digestion leads to a delay in the production of milk proteins as resources from the blood meal are not yet available. Importantly, the consumption of blood at suboptimal temperatures leads to a slight reduction in population growth, suggesting that tsetse fly maximum reproductive output requires feeding on warmer blood.

In conclusion, we provide evidence that consumption of warm blood meals may not be purely detrimental to blood feeding insects as previously observed in other systems. Rather the impact is species specific and can be nuanced, as specific aspects may be beneficial for the insect physiology, such as the processing of the blood meal and reproductive output in tsetse flies.

## Figures and Tables

**Figure 1 insects-13-00997-f001:**
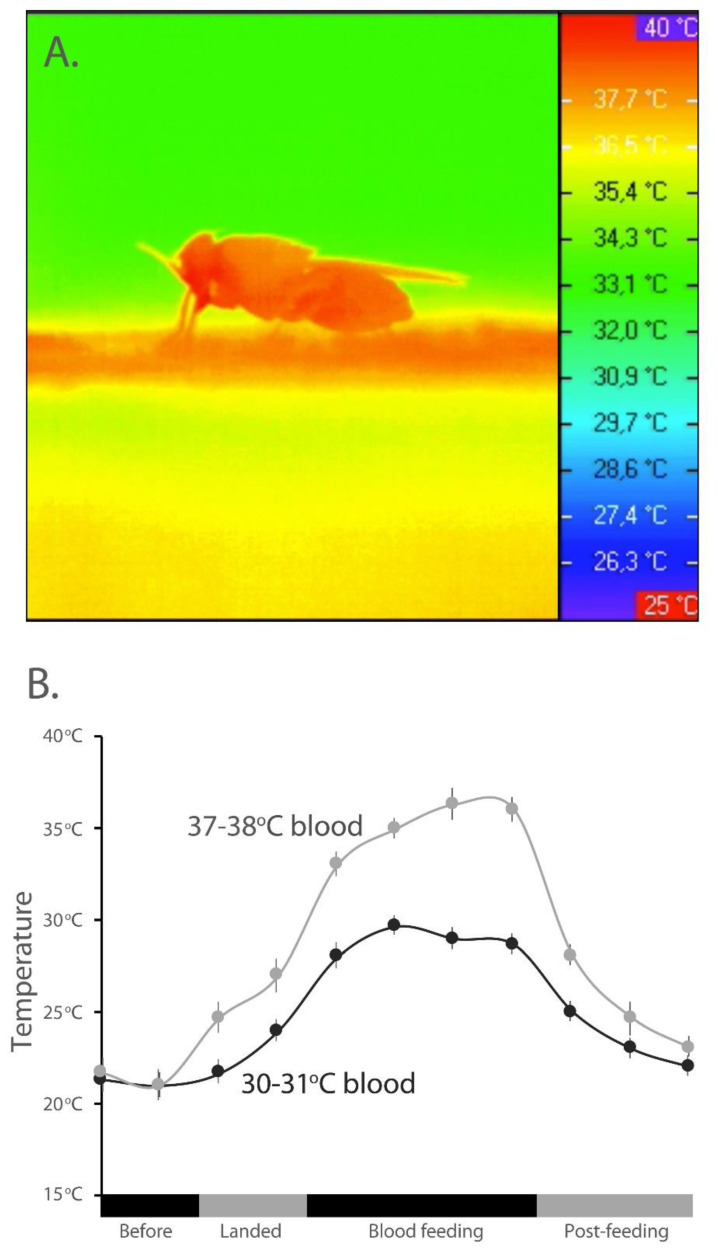
**Thermal changes in tsetse flies during the consumption of blood.** (**A**). Thermal image of tsetse fly when feeding on warm blood. (**B**). Thermal changes in tsetse fly abdomen when consuming warm and cool blood with the use of a thermocouple. There is a significant reduction in temperature from when the fly landed on the host until after the completion of feeding.

**Figure 2 insects-13-00997-f002:**
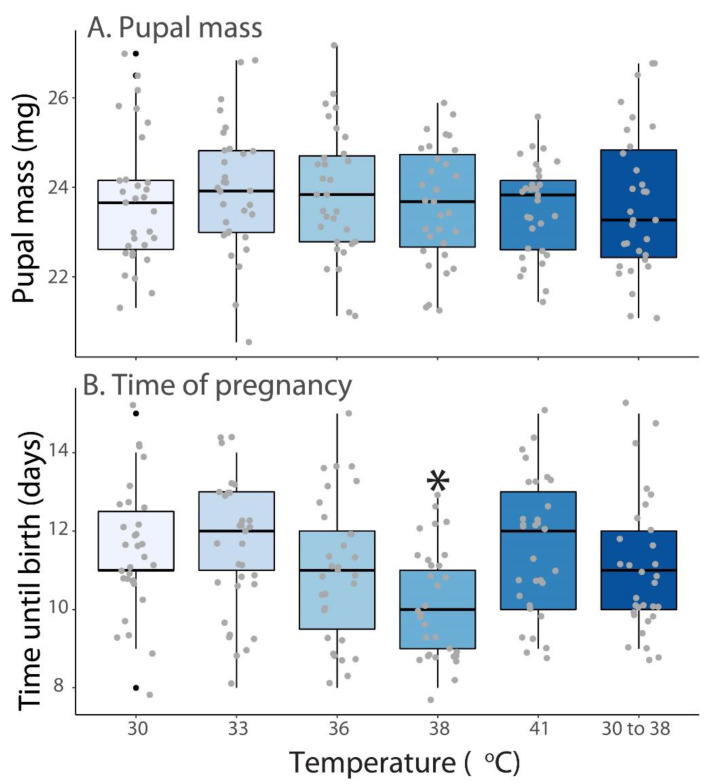
**Progeny size and pregnancy duration changes in relation to blood meal temperature.** (**A**). Pupal mass and (**B**). time of pregnancy in relation to the composition of blood at multiple temperatures. 30 to 38 °C represents a group of flies provided a cool bloodmeal for two feeding periods and a warm bloodmeal for the other two. Statistical differences were assessed with the use of ANOVA followed by post hoc tests. Different colors represent temperature differences. Gray dots indicate the range of pupal mass and time of pregnancy. Black dots indicate extreme values. Asterisks indicate significant differences.

**Figure 3 insects-13-00997-f003:**
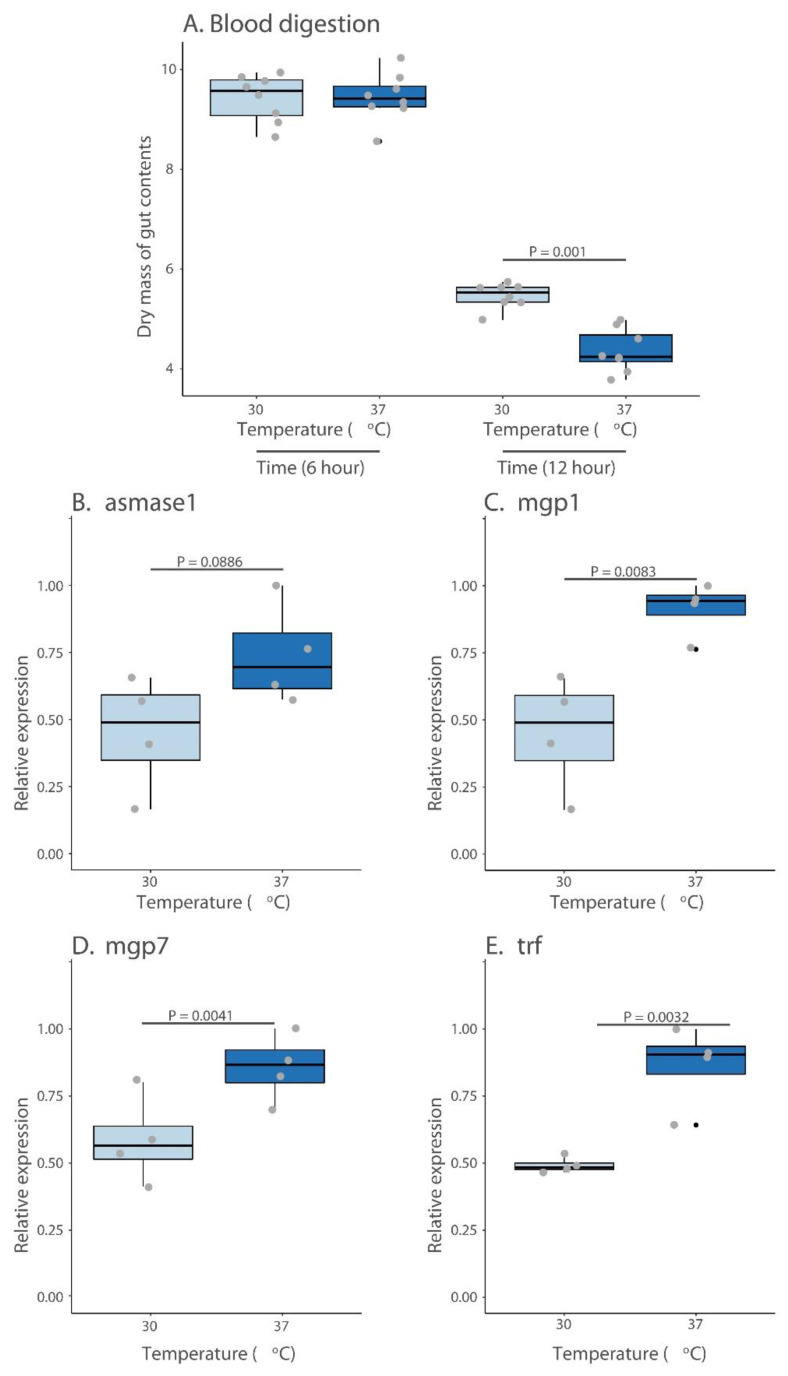
**Blood digestion and milk gland protein transcript abundance.** (**A**). Dry mass of the *G*. *morsitans* midgut 6 and 12 h following a blood meal. (**B**–**E**). Transcript levels for four milk gland protein when flies consumed cool (30 °C) or warm blood (37–38 °C). Data is displayed as mean. Statistical differences were assessed with the use of a Student *t*-test or Welch’s test for non-normal data (results were similar for both tests). Different colors represent temperature differences. Gray and black dots indicate the non-normal data.

**Figure 4 insects-13-00997-f004:**
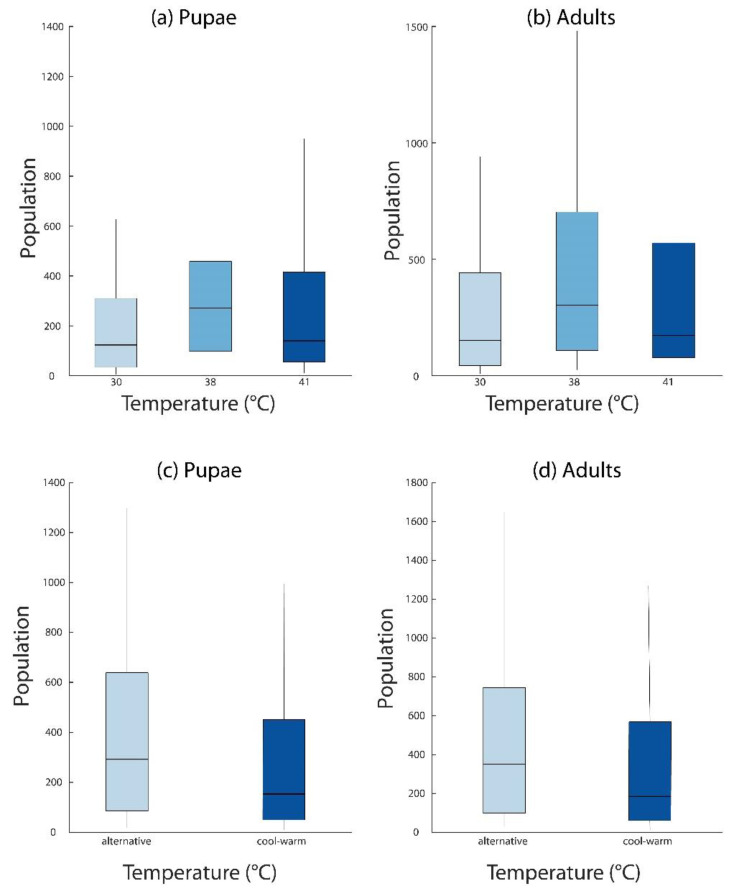
**Population modeling based on changes in pregnancy duration following feeding.** (**a**,**b**). Pupae and adult population levels when consuming cool, warm, and hot blood. Values for all temperatures are provided in Appendix A. (**c**). A cool blood meal (30 °C) followed by a second warm blood meal (38 °C) each week still leads to a lower population growth than those offered a warm blood meal (38 °C) for all meals (**d**).

## Data Availability

Data is contained within the article or Appendix A.

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
