# Peer review of "Warm Blood Meal Increases Digestion Rate and Milk Protein Production to Maximize Reproductive Output for the Tsetse Fly, Glossina morsitans"

_insects, 2022, doi:10.3390/insects13110997_

Round 1

Reviewer 1 Report

Comments to the manuscript

The manuscript is well written, and it was easy to follow the reasoning in the introduction and the discussion. The difficult part of this text is the M&M section and its connection to the RESULTS section. I feel that these parts need changes and a somewhat better structure. It was hard to understand exactly what had been done and how the measurements are conducted.

L22: period for arthropod vectors

L26: flies not files

L27-28: increased growth rates ??? – is there an option to warm bloodmeals??? – see also some later comments…

L30: perhaps use a different word than pressure… burden??

INTRODUCTION

The introduction starts with vector control, but do not follow this up later in the text. This is probably difficult because most bloodmeals taken will be in the range of 36-38°C. I might be mistaken on this part, but I feel that the beneficial effect is mostly a theoretical aspect which might not occur often under natural condition... ? The findings are still interesting because it shows that the short temperature stress is a complex situation which also may give some benefits. The introduction is also quite short and very to the point. I believe it would improve the manuscript if the introduction could be expanded on the current knowledge of stress/benefit in a slightly larger context. I guess benefits are poorly studied even though several blood feeding diptera species acts as pests or vectors. This can be pointed out. Highlighting these aspects can broaden the interest for these kinds of studies. It might also be room for some expanded information on differences, or potential differences, between tsetse and blood feeders with normal egg deposition…

L48-50: Add references

M & M

This part needs improvement. I really appreciate that many of the technical parts are tucked away in previous publications, but there is too little information left. It was impossible for me to grasp how the different experiments are conducted and there is no description on the various treatments. A description of the experimental progress needs to be included. As it is now the M&M section consist of fragments. Examples: Which temperatures are used (treatments does not appear until the results)? How many flies were used? How and when were flies for gut examination collected/selected? When and at what temperatures were blood given (in the results it seems that some flies have received two cold meals)? Did you follow cohorts and removed some of the flies and let other produce offspring? how was pupae measured? … The M&M section clearly needs more information. It is difficult to figure out what has been measured and why …

I also feel that there should be better consistency between the paragraph numbering (2.1, 2.2, 2.3… etc.) in relation to the results (3.1, 3.2, 3.3… etc.)

L83: Add G. morsitans at the beginning of the sentence

L87-96: Two methods are described (imaging and thermocouple) … which was used in analyses/results… perhaps averages of the two?

L93: add reference on thermocouple

L104: reference in wrong format

Is the three paragraphs 2.3, 2.4 and 2.5 all directly connected to the results in 3.3?

L116: I guess primers were used in this study as well ?

L122-145: a suggestion: the population modeling chapter can be written in a way that explains the parameters, why they are used and why they are of interest for population dynamics. It is not intuitive for all readers how this model works. The more “technical” part can then be moved to a paragraph with “Statistics and modeling parameters”. In this new “statistics-part” different tests can also be explained and the triple mention of R-packages (L96, L105, L120) can be avoided. 

RESULTS

Figure 1 has temperatures 30-31°C and 37-38°C, Figure 2 has temperatures 30°C, 33°C, 36°C, 38°C, 41°C and 30-38°C and figure 3 has temperatures 30°C and 37°C … This is confusing and needs an explanation in M&M… what has been done and what has been tested?

Figure 1:              No measures in between the two temperatures?

Figure 2: I don’t understand the pooling of 30-38°C? I guess 36°C and 38°C are within the range of expected bloodmeal temperature for this this species and the other temperatures are above and below expected temperatures 

L161-162: This text resemble discussion and can be moved there… ?

L176-178: cool bloodmeal twice… compared to what - a hot bloodmeal twice or individuals that consumed a warmer meal once… Figure legend indicates “a bloodmeal”. This is why a proper description of the progress of the experiment is needed

L176-178: This text resemble discussion and can be moved there… ?

Figure 4: all y-axis should be equal.

DISCUSSION

I guess the observations in this study is an effect of artificial laboratory feeding at suboptimal temperatures. This should be pointed out and discussed. Do Tsetse feed on blood sources outside of 36-38°C? Are there any options… If so… at what extent do these options exist?

I also believe that if the introduction is expanded with more information on known knowledge gaps or other diptera pests and vectors this can be picked up in the discussion and potentially be related to your findings. If benefits from elevated temperatures is an understudied aspect this should be pointed out clearly. The conclusion is interesting as it highlights the complexity of physiological responses and would be even more interesting if put in a context of other relevant species. Even if cold bloodmeals should be infrequent under natural condition the findings of this study are relevant for understanding the physiological mechanisms behind population development and may therefore indirectly contribute to potential vector control.       

L207: … reproductive output …

Author Response

Review 1

  1. Comment: The manuscript is well written, and it was easy to follow the reasoning in the introduction and the discussion. The difficult part of this text is the M&M section and its connection to the RESULTS section. I feel that these parts need changes and a somewhat better structure. It was hard to understand exactly what had been done and how the measurements are conducted.

Response: Thanks for the comments.  I have addressed those identified below.

  1. Comment: L22: period for arthropod vectors

Response: This has been changed as suggested.

  1. L26: flies not files

Response: This has been changed as suggested.

  1. L27-28: increased growth rates ??? – is there an option to warm bloodmeals??? – see also some later comments…

Response: We have added that this is in comparison to cool bloodmeals.

  1. L30: perhaps use a different word than pressure… burden??

Response: This has been changed to burden.

  1. The introduction starts with vector control, but do not follow this up later in the text. This is probably difficult because most bloodmeals taken will be in the range of 36-38°C. I might be mistaken on this part, but I feel that the beneficial effect is mostly a theoretical aspect which might not occur often under natural condition... ? The findings are still interesting because it shows that the short temperature stress is a complex situation which also may give some benefits. The introduction is also quite short and very to the point. I believe it would improve the manuscript if the introduction could be expanded on the current knowledge of stress/benefit in a slightly larger context. I guess benefits are poorly studied even though several blood feeding diptera species acts as pests or vectors. This can be pointed out. Highlighting these aspects can broaden the interest for these kinds of studies. It might also be room for some expanded information on differences, or potential differences, between tsetse and blood feeders with normal egg deposition…

Response: A. We have deleted the section on vector control, as we agree that it is not important. B. We have expanded the section on heat stress and feeding. C. We agree that a section on oviparous vs. viviparous would be interesting, but would be out of the scope of this study and would be more set for a review. We did add a bridge sentence that highlights tsetse flies are different.

  1. L48-50: Add references

Response: This section has been deleted.

  1. This part needs improvement. I really appreciate that many of the technical parts are tucked away in previous publications, but there is too little information left. It was impossible for me to grasp how the different experiments are conducted and there is no description on the various treatments. A description of the experimental progress needs to be included. As it is now the M&M section consist of fragments. Examples: Which temperatures are used (treatments does not appear until the results)? How many flies were used? How and when were flies for gut examination collected/selected? When and at what temperatures were blood given (in the results it seems that some flies have received two cold meals)? Did you follow cohorts and removed some of the flies and let other produce offspring? how was pupae measured? … The M&M section clearly needs more information. It is difficult to figure out what has been measured and why …

Response: We have added additional details as requested throughout the methods.

  1. I also feel that there should be better consistency between the paragraph numbering (2.1, 2.2, 2.3… etc.) in relation to the results (3.1, 3.2, 3.3… etc.)

Response: Specific method sections cannot be merged to match the results. We had an earlier draft in this format and it was more confusing. We prefer to leave in the current format. 

  1. L83: Add  morsitans at the beginning of the sentence

Response: This has been corrected.

  1. L87-96: Two methods are described (imaging and thermocouple) … which was used in analyses/results… perhaps averages of the two?

Response: We have added details for clarification.

  1. L93: add reference on thermocouple

Response: We have added this reference.

  1. L104: reference in wrong format

Response: This has been corrected.

  1. Is the three paragraphs 2.3, 2.4 and 2.5 all directly connected to the results in 3.3?

Response: Yes.

  1. L116: I guess primers were used in this study as well ?

Response: We have edited for clarity.

  1. L122-145: a suggestion: the population modeling chapter can be written in a way that explains the parameters, why they are used and why they are of interest for population dynamics. It is not intuitive for all readers how this model works. The more “technical” part can then be moved to a paragraph with “Statistics and modeling parameters”. In this new “statistics-part” different tests can also be explained and the triple mention of R-packages (L96, L105, L120) can be avoided. 

Response: We have moved triple mention to

  1. Figure 1 has temperatures 30-31°C and 37-38°C, Figure 2 has temperatures 30°C, 33°C, 36°C, 38°C, 41°C and 30-38°C and figure 3 has temperatures 30°C and 37°C … This is confusing and needs an explanation in M&M… what has been done and what has been tested?

Response: We have added more details Figure 1 was set at 30°C, but increased to 31°C (37°C to 38°C). The others were held at a single temperature. We have added the temperature could vary 1°C.

  1. Figure 1: No measures in between the two temperatures?

Response: No, we were only highlighting the differences and used the two temperatures.

  1. Figure 2: I don’t understand the pooling of 30-38°C? I guess 36°C and 38°C are within the range of expected bloodmeal temperature for this this species and the other temperatures are above and below expected temperatures 

Response: We have added info that the treatment was flies first provided a cool meal and then warm meals.

  1. L161-162: This text resemble discussion and can be moved there… ?

Response: This is a brief statement that we would like to leave here.

  1. L176-178: cool bloodmeal twice… compared to what - a hot bloodmeal twice or individuals that consumed a warmer meal once… Figure legend indicates “a bloodmeal”. This is why a proper description of the progress of the experiment is needed

Response: We have described this in more details in the methods section. This should help to clarify.

  1. L176-178: This text resemble discussion and can be moved there… ?

Response: This is a short summary. We would prefer to leave here.

  1. Figure 4: all y-axis should be equal.

Response: The pupae axis are the same, but we prefer to leave the current y-axis.

  1. I guess the observations in this study is an effect of artificial laboratory feeding at suboptimal temperatures. This should be pointed out and discussed. Do Tsetse feed on blood sources outside of 36-38°C? Are there any options… If so… at what extent do these options exist?

Response: We have added info that this is a possibility to the discussion.

  1. I also believe that if the introduction is expanded with more information on known knowledge gaps or other diptera pests and vectors this can be picked up in the discussion and potentially be related to your findings. If benefits from elevated temperatures is an understudied aspect this should be pointed out clearly. The conclusion is interesting as it highlights the complexity of physiological responses and would be even more interesting if put in a context of other relevant species. Even if cold bloodmeals should be infrequent under natural condition the findings of this study are relevant for understanding the physiological mechanisms behind population development and may therefore indirectly contribute to potential vector control.      

Response: We have expanded the introduction and discussion.

  1. L207: … reproductive output …

Response: Fixed as suggested

Reviewer 2 Report

Overall, an interesting report showing that, differently from other blood-feeding insects where body temperature after a hot blood meal was actively decreased, there is no temperature-lowering mechanism acting in the tsetse fly. Conversely, heating seems to be turned into a homeostatic stimulus. The paper is descriptive, with few mechanistic data or insights. As someone interested in adaptations to hematophagy, I finished the article with a long list of hypotheses and experiments. But this report, although simple, provides a good contribution to the field and can stand as it is. The paper is written in a straightforward way and reads well. I have only two general comments and a few specific observations.

General comments

This insect behaves differently from other hematophagous insects: it does not need to prevent heating but instead relies on it. This simple fact should be more stressed from the beginning. Only at the end of the discussion do the authors state that “as specific aspects may be beneficial for the insect physiology, such as the processing of the blood meal and reproductive output in tsetse flies.” Therefore, it stays without being mentioned the most exciting question for future research: How? This would call for mechanistic insights that are missing in the paper. Some could be addressed experimentally (Protease activity? Intestinal motility?). I guess that if heat is not a problem, then some peculiarity of the stress response in this fly is rendering the insect tolerant to a heat shock, possibly by faster or higher expression of stress proteins. But, of course, this may sound too speculative.

What is the meaning of population modeling? The results from figures 1-3 clearly show that heating is a physiological stimulus rather than a problem as has been postulated for other species. However, in a field situation, there would be no feeding on cold blood. The modeling is not wrong, so the authors can stay with that if they like, but it does not seem to improve the major conclusions of the paper.

Minor comments

Line 25 – files> flies

Line 116 - All readings were obtained on four biological replicates – samples used for each of the replicates were from single flies?

Line 126 – duration means duration of the developmental stage?

Line 127 – define developmental rate

Line 131 – dA1 should be dA2, no?

Figure 1B shows several points collected at presumably different time intervals as they are shown sequentially, but only the phases of the feeding process are shown. Please insert a time scale or provide this information in some way.

In this experiment, the amount of blood ingested was the same in both temperatures? The number of insects and replicates is not provided.

Figure 2 B – Do samples pass on a normality test? Data shown are mean or median?  

Figure 3 - it is unclear when milk gland protein expression samples were collected. What is the duration of blood meal digestion in the fly? This information would help people that are not specialists in glossina biology.

Also, define the term larval “deposition” for a non-glossina person.

Line 224 – something is missing in the sentence. Please complete.

Author Response

  1. Overall, an interesting report showing that, differently from other blood-feeding insects where body temperature after a hot blood meal was actively decreased, there is no temperature-lowering mechanism acting in the tsetse fly. Conversely, heating seems to be turned into a homeostatic stimulus. The paper is descriptive, with few mechanistic data or insights. As someone interested in adaptations to hematophagy, I finished the article with a long list of hypotheses and experiments. But this report, although simple, provides a good contribution to the field and can stand as it is. The paper is written in a straightforward way and reads well. I have only two general comments and a few specific observations.

Response:  Thanks you for the comments.

  1. This insect behaves differently from other hematophagous insects: it does not need to prevent heating but instead relies on it. This simple fact should be more stressed from the beginning. Only at the end of the discussion do the authors state that “as specific aspects may be beneficial for the insect physiology, such as the processing of the blood meal and reproductive output in tsetse flies.” Therefore, it stays without being mentioned the most exciting question for future research: How? This would call for mechanistic insights that are missing in the paper. Some could be addressed experimentally (Protease activity? Intestinal motility?). I guess that if heat is not a problem, then some peculiarity of the stress response in this fly is rendering the insect tolerant to a heat shock, possibly by faster or higher expression of stress proteins. But, of course, this may sound too speculative.

Response: It is very likely that factors underlying digestion are more rapid, such as protease activity. We don’t think that the heat shock protein expression is higher, rather they are likely underlying more tolerant. We have added reference to this in the discussion, but don’t want to speculate on the specific mechanism.  

  1. What is the meaning of population modeling? The results from figures 1-3 clearly show that heating is a physiological stimulus rather than a problem as has been postulated for other species. However, in a field situation, there would be no feeding on cold blood. The modeling is not wrong, so the authors can stay with that if they like, but it does not seem to improve the major conclusions of the paper.

Response: We have added info to the discussion that they could easily feed on cooler bloodmeals as temperature can vary. This is now included in the discussion.

  1. Line 25 – files> flies

Response: Fixed as suggested.

  1. Line 116 - All readings were obtained on four biological replicates – samples used for each of the replicates were from single flies?

Response: We have added that two flies were merged per replicate.

  1. Line 126 – duration means duration of the developmental stage?

Response: We have clarified.

  1. Line 127 – define developmental rate

Response: We have clarified.

  1. Line 131 – dA1 should be dA2, no?

Response: Yes, this has been corrected.

  1. Figure 1B shows several points collected at presumably different time intervals as they are shown sequentially, but only the phases of the feeding process are shown. Please insert a time scale or provide this information in some way.

Response: We have added info the figure legend.

  1. In this experiment, the amount of blood ingested was the same in both temperatures? The number of insects and replicates is not provided.

Response: We believe so, as there were no differences after 6 hours after a bloodmeal.

  1. Figure 2 B – Do samples pass on a normality test? Data shown are mean or median?  

Response: We have clarified this in the figure legend.

  1. Figure 3 - it is unclear when milk gland protein expression samples were collected. What is the duration of blood meal digestion in the fly? This information would help people that are not specialists in glossina biology.

Response: We have added clarification to the methods.

  1. Also, define the term larval “deposition” for a non-glossina person

Response: We have defined as birth.

  1. Line 224 – something is missing in the sentence. Please complete.

Response: We have fixed.